# Neuroacoustic Patterns: Constant Q Cepstral Coefficients for the Classification of Neurodegenerative Disorders

## Abstract

Early identification of neurodegenerative diseases is crucial for effective diagnosis in neurological disorders. However, the quasi-periodic nature of vocal tract sampling often results in inadequate spectral resolution in traditional spectral features, such as Mel Frequency Cepstral Coefficients (MFCC), thereby limiting their classification effectiveness. In this study, we propose the use of Constant Q Cepstral Coefficients (CQCC), which leverage geometrically spaced frequency bins to provide superior spectrotemporal resolution, particularly for capturing the fundamental frequency and its harmonics in speech signals associated with neurodegenerative disorders. Our results demonstrate that CQCC, when integrated with Random Forest and Support Vector Machine classifiers, significantly outperform MFCC, achieving absolute improvements of 5.6 % and 7.7 %, respectively. Furthermore, CQCC show enhanced performance over traditional acoustic measures, such as Jitter, Shimmer, and Teager Energy. The effectiveness of CQCC is underpinned by the form-invariance property of the Constant Q Transform (CQT), which ensures consistent feature representation across varying pitch and tonal conditions, thereby enhancing classification robustness. Furthermore, the robustness of CQCC features against MFCC features are validated using LDA plots. These findings are validated using the Italian Parkinson's database and the Minsk2019 database of Amyotrophic Lateral Sclerosis, underscoring the potential of CQCC to advance the classification of neurodegenerative disorders.

## 1 Introduction

Neurodegenerative disorders have become a significant and escalating health concern as populations age globally. These disorders are marked by the gradual loss of neuronal function, leading to debilitating cognitive and motor impairments. Despite the availability of advanced medical technologies, diagnosing and managing these diseases remain significant challenges. The complexities of these conditions, coupled with the limitations of current therapies, emphasize the urgent need for innovative approaches to both diagnosis and treatment.

Neurodegeneration stands out as the central pathological process in the majority of brain-related conditions (Jeong et al., 2024). Conditions like Parkinson's Disease (PD) and Amyotrophic Lateral Sclerosis (ALS) continue to be major clinical challenges, especially within the elderly demographic. (Garofalo et al., 2020). The World Health Organization's report on Neurological Disorders: Public Health Challenges indicates that nearly one billion people worldwide are affected. (Bosco et al., 2011). The formidable blood-brain barrier (BBB) continues to pose a major challenge in the effective management of neurodegenerative disorders (NDs). The WHO has noted that, despite the availability of highly effective and affordable treatments, up to 9 out of 10 individuals with NDs in developing countries remain untreated. Enhancing health systems is essential to provide better care for those with neurological disorders. Despite ongoing efforts in modern science to develop medical or surgical interventions, the results have been largely disappointing. This underscores the critical need for further research in this field.

Language deficits are frequently observed in numerous neurodegenerative conditions, often emerging early as a prominent symptom. Therefore, identifying and characterizing language impairments

in patients with NDs is becoming increasingly important for diagnosing various neurodegenerative diseases. (Boschi et al., 2017). Furthermore, neurodegenerative disorders can impact speech due to the decline in motor control. Symptoms of PD related to the motor system include tremors, rigidity, poor balance, and slow movement. (Jeong et al., 2024).Specifically, motor speech irregularities associated with PD impact elements such as prosody, resonance, articulation, breathing, and phonation.(Magee et al., 2019).Although the exploration of language-related issues in Amyotrophic Lateral Sclerosis (ALS) has been limited, some studies have highlighted language deficits in ALS patients without dementia, revealing the presence of diverse cognitive profiles. Individuals with ALS may face challenges with articulation and understanding sentence structure, resulting in simplified syntax and difficulties in comprehending complex syntax. This motivates researchers to develop diagnostic assistive speech tools to aid in the classification of various Neurodegenerative Diseases (NDs). The literature predominantly focuses on the classification of PD in comparison to ALS.

## 2 RELATED WORK

In recent decades, there has been increasing interest in automatically identifying neurological diseases through the analysis of vocal recordings.(Benba et al., 2016; Rusz et al., 2011; Orozco-Arroyave et al., 2016). In the study referenced in (Kim, 2017), the authors examine the fricative sounds produced by individuals with Parkinson's Disease (PD). The study also explores the significance of nasal consonants in the automatic identification of PD.(Spangler et al., 2017). In (Moro-Velazquez et al., 2019), the role of nasal consonants in the automatic identification of individuals with Parkinson's Disease (PD) was explored. In (Moro-Velazquez et al., 2019), the authors also proposed a method utilizing Perceptual Linear Prediction (PLP) features and Gaussian Mixture Models (GMM) with Universal Background Models (UBM) classifiers for the classification of Healthy vs. PD.

In (Vashkevich & Rushkevich, 2021), various acoustic features such as Jitter, Shimmer, Mel Frequency Cepstral Coefficients (MFCC), Formant Frequencies, and Pitch Period Entropy are used for the classification of Healthy individuals vs. those with neurodegenerative disease based on sustained vowels. Additionally, (Simmatis et al., 2024) also utilized acoustic and articulatory features for ALS classification. However, there is limited research on classifying multiple neurodegenerative diseases simultaneously. The study reported in (Suhas et al., 2020) investigates a Mel-Spectrogram-based approach for distinguishing between Parkinson's Disease, ALS, and Healthy Controls. Aditionally, in the context of Heisenberg's uncertainty principle applied to signal processing, the Short-Time Fourier Transform (STFT) used in MFCC imposes a fixed time-frequency resolution across the entire time-frequency plane. Moreover, it lacks the *form-invariance* property, as the analysis window in STFT depends exclusively on the *time* parameter(Gambardella, 1968). To that effect, we propose a novel feature extraction method based on the Constant-Q Transform (CQT) and its cepstral representation, known as Constant Q Cepstral Coefficients (CQCC) for classification of Neurodegenerative disorders. Originally introduced in the context of antispoofing literature (Todisco et al., Bilbao, Spain, June 21-24, 2017), CQCC has also demonstrated strong performance in the classification of pathological infant cries (Patil et al., 2023).

●Furthermore, no studies have reported on capturing the neurodegenerative disease on sustained vowel sounds through *Form-Invariance* property of CQT.

● To the best of the authors' knowledge, this is the first study of it;s kind on sustained vowel sounds for multi neurodegenerative disorder classification and analysis.

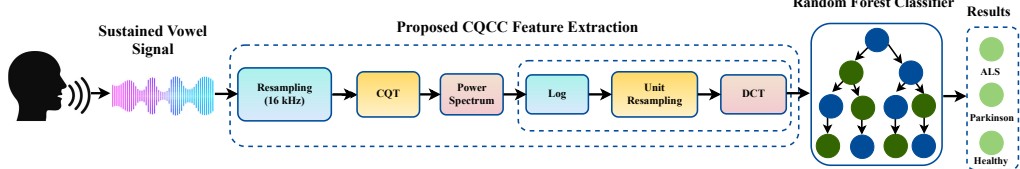

Figure 1: Functional block diagram of proposed CQCC Feature Set for Classification of Neurodegenerative Disorders. After (Patil et al., 2023). Best viewed in colour.

Table 1: Window length $\mathsf{v}(m, r)$ in samples as a function of analysis frequency ($g_r$). Adapted from (Brown, 1991).

| r | Frequency ($Hz$) | # Samples | Duration ( in $ms$) |
|---|---|---|---|
| 1 | 100 | 29547 | 1340 |
| 100 | 204.37 | 14457 | 655.64 |
| 200 | 420 | 7022 | 318.48 |
| 400 | 1783 | 1657 | 75.15 |
| 600 | 7556 | 391 | 17.73 |

## 3 METHODOLOGY

This section discusses about the Constant Q transform (CQT) feature extraction and the *Form-Invariance* property of CQT

### 3.1 THE CONSTANT-Q TRANSFORM (CQT)

The Discrete Fourier Transform (DFT) is essentially a sampled version of the Discrete-Time Fourier Transform (DTFT) applied to each frame of the speech signal (Brown, 1991). Let $z(m)$ be the discrete-time input speech signal with a sampling rate of $F_r$. The Short-Time Fourier Transform (STFT) of $z(m)$ is expressed as (Quatieri, 2015):

$$Z(\theta, \mu) = \sum_{m=-\infty}^{\infty} z(m) \cdot \mathsf{v}(m, \mu) \cdot e^{-j\theta m}, \tag{1}$$

where $\mathsf{v}(m, \mu)$ denotes the analysis window centered at time $\mu$. It is important to note that $\mathsf{v}(m, \mu)$ is a function of only the time variable $\mu$. Furthermore, let $w_p(m) = z(m)\mathsf{v}(m, \mu)$ represent a windowed frame of the speech signal, then the $M$-point DFT, $W_p(r)$, of $w_p(m)$ can be represented as:

$$W_p(r) = \sum_{m=0}^{M-1} w_p(m) \cdot e^{-j(\frac{2\pi}{M})rm}, \tag{2}$$

where $r$ is the frequency bin index, and $\theta_{DFT} = (2\pi r)/M$ (i.e., uniform frequency spacing). In this research, we have employed the CQCC instead of the STFT-based feature sets. The Constant-Q Transform (CQT) offers superior frequency resolution in lower frequency regions. In CQT, the quality factor $P$ of the subband filters used in the filter bank remains constant (as discussed in eq. (5)), thus leading to geometrically spaced frequency bins as introduced in Brown's original work (Brown, 1991). The CQT of a signal $w_p(m)$ is given by:

$$W_p^{CQT}(r) = \frac{1}{M(r)} \sum_{r=0}^{M(r)-1} w_p(m)\mathsf{v}(m, r)e^{-j\left(\frac{2\pi}{M(r)}Pm\right)}, \tag{3}$$

where $\theta_{CQT} = (2\pi Pm)/M(r)$, and $\mathsf{v}(m, r)$ is the analysis window, which has a consistent shape for the analysis of each frequency component $g_r$, though its length is determined by $M(r)$, making it a function of both time $(m)$ and frequency $(r)$, where

$$M(r) = P\left(\frac{F_r}{g_r}\right). \tag{4}$$

It should be observed that $\mathsf{v}(m, \mu)$ in eq. (1) is only a function of the time parameter '$\mu$', whereas $\mathsf{v}(m, r)$ in eq. (3) is a function of both time $(m)$ and frequency $(r)$. Table 1 displays the window durations for the CQT parameter set in infant cry classification. From Table 1, it can be seen that the window length varies with respect to $g_r$, reducing as $g_r$ increases. The window duration is significantly larger in the lower frequency regions, offering high frequency resolution, making the CQT an effective method to capture infant cry characteristics in lower frequency ranges.

**Algorithm 1:** Pseudo-Code of the Revised CQCC Feature Set. Adapted from (Patil et al., 2023).

1: $g_m = (2^{\frac{m-1}{D}})g_{min}$                                *geometrically spaced frequency bins*

2: $M(m) = \frac{S_r}{\Delta g_m}$

3: $Z_r^{CQT}(m) = \left\langle z_r(p) \cdot \psi(p,m), e^{j\frac{2\pi Rp}{M(m)}} \right\rangle$    *computation of CQT for the speech segment $z_r(m)$*

4: **for** $j = 1 : N_{columns}(Z_{CQT})$ **do**

5:      Framewise concatenation of CQT:

6:      $Z_{CQT}(m,j) = Z_{z_{rj}}^{CQT}(m)$   *CQT computed for the corresponding segment $z_{r_j}(p)$ for the $j^{th}$ column*

7: **end for**

8: $Z_{CQT}^{resampled}(m,j) = resample(Z_{CQT}(m,j))$     *Frequency bins resampled for linear spacing*

9: $CQCC = DCT\left(\log\left(Z_{CQT}^{resampled}(m,j)\right)\right)$

Since the quality factor $P$ is the ratio of center frequency to bandwidth, it is defined as (Brown, 1991):

$$P = \frac{g_r}{\Delta g_r} = \frac{g_r}{g_{r+1} - g_r} = \frac{1}{2^{1/B} - 1}, \tag{5}$$

where $B$ is the number of bins per octave, and $g_r$ represents the frequency of the $r^{th}$ spectral component, as defined by (Brown, 1991):

$$g_r = (2^{(r-1)/B})g_{min}, \tag{6}$$

where $g_{min}$ is the minimum frequency of the signal. Additionally, we resampled the magnitude spectrum of the CQT to a linear scale to reduce the number of frequency bins in the feature set (Todisco et al., Bilbao, Spain, June 21-24, 2017). Substituting eq. (5) into eq. (4), we have:

$$M(r) = \frac{F_r}{\Delta G_r}. \tag{7}$$

Additionally, we converted the geometrically-spaced frequency scale to a linearly-spaced one to maintain the orthogonality of the Discrete Cosine Transform (DCT). Since frequency bins in CQT are geometrically spaced, reconstructing the signal can be viewed as a downsampling operation for the initial $r$ bins, corresponding to lower frequencies, and as upsampling for the remaining $R - r$ bins, corresponding to higher frequencies. Further details on resampling can be found in (Todisco et al., Bilbao, Spain, June 21-24, 2017). Applying the DCT to the resampled CQT produces the CQCC feature set. The pseudo-code for CQCC feature extraction is given in Algorithm 1. Figure 1 outlines the functional block diagram of the proposed CQCC-based neurodegenerative disease classification system.

### 3.1.1 FORM-INVARIANCE PROPERTY OF CQT

For simplicity, we examine the continuous-time forms of the Fourier Transform (FT), Short-Time Fourier Transform (STFT), and Constant-Q Transform (CQT). If $y(t)$ and $Y(\xi)$ are a Fourier transform pair, then the time-scaling property of the Fourier Transform can be expressed as follows (Gambardella, 1968), (Quatieri, 2015):

$$\mathcal{F}\{y(\beta t)\} = \frac{1}{|\beta|}Y\left(\frac{\xi}{\beta}\right), \tag{8}$$

indicating that scaling the time domain by a factor of $\beta$ corresponds to scaling the frequency domain by the *inverse* factor $\frac{1}{\beta}$. This shows that the structure of the energy spectral density (ESD) remains unchanged, which is why this property is referred to as "form-invariance." However, this property does not extend to the conventional STFT, where the analysis window function depends solely on the time variable.

Schroeder and Atal introduced the STFT using practically realizable bandpass Linear Time-Invariant (LTI) filters (Schroeder & Atal, 1962), defining it as follows:

$$Y(t, \xi) = \int_{-\infty}^{t} y(\theta) \psi(t - \theta) e^{-j\xi\theta} d\theta. \tag{9}$$

For the STFT to be form-invariant, the following condition must hold:

$$STFT\{y(lt)\} = Y_l(t, \xi) = \eta Y(\gamma t, \delta \xi), \tag{10}$$

where $\gamma$ and $\delta$ represent time and frequency scaling factors, respectively. It has been shown that realizing this condition places necessary and sufficient constraints on the window function, requiring it to belong to a class of single-term power functions: $\psi(t) = ct^d$, $t > 0$, where $c$ and $d$ are real constants. According to the stability condition for LTI filters, this window is *unstable*, making it impractical for real-world applications.

However, it is interesting to note that the situation changes if the window function depends on both time and frequency, i.e., $\psi(t) \equiv \psi(t, \xi)$, as in the case of CQT. In this scenario, the STFT equation becomes the following:

$$Y(t, \xi) = \int_{-\infty}^{t} y(\theta) \psi(t - \theta, \xi) e^{-j\xi\theta} d\theta, \tag{11}$$

and the form-invariance condition is satisfied for the window function. Further technical details of this condition are provided in the Appendix. Specifically, the window function takes the following form:

$$\psi(t, \xi) = cv(t\xi)t^d, \quad t > 0, \, l > 0, \, \xi > 0, \tag{12}$$

where $v(t\xi)$ is an arbitrary real function of $t\xi$, and $c$ and $d$ are real constants. Furthermore, $\psi(t, \xi)$ also adheres to the Bounded Input and Bounded Output (BIBO) stability conditions for an LTI filter, meaning that its impulse response is absolutely integrable (Oppenheim et al., 2001), as expressed in the following condition:

$$\int_{-\infty}^{+\infty} |\psi(t, \xi)| dt < \infty. \tag{13}$$

Moreover, this form of the window function applies to practical models involving short-time analysis, such as those that mimic the auditory system's peripheral processing. For example, Flanagan's original model (Flanagan, 2013) describes the window function used in mechanical spectral analysis due to the movements of the basilar membrane in the cochlea of the human ear (Gambardella, 1968). In particular, the window function is given by $\psi(t, \xi) = (t\xi)^2 e^{-\frac{t\xi}{2}}$, which is similar to the form described above.

## 4 EXPERIMENTAL SETUP

### 4.1 DATASETS DETAILS

In this study, we use Italian Parkinson's Voice and Speech dataset, which was designed in accosication with "Associazione Parkinson Puglia" (Dimauro & Girardi, 2019). Aditionally, we also used Minsk2019 ALS database which was designed using recordings made from Republican Research and Clinical Center of Neurology and Neurosurgery (Minsk, Bela) (Vashkevich et al., 2019). Both dataset contains the sustained sounds of all vowel sounds. Since the sampling rate of the cry signals provided in the dataset is not uniform, we resampled all the utterances at a sampling rate of 16 kHz. The dataset consists of sustained vowel phonations from individuals diagnosed with Parkinson and ALS along with healthy controls at a comfortable pitch and loudness as constant and long as possible. The imabalce in dataset was handled using *SMOTE*. For training and testing, we used $80\%$ and $20\%$ of the data, respectively. Table 2 shows the statistics of all datasets utilized. Further, 3, shows the agewise distribution of parkinson patients and healthy controls in the Italian Parkinson Dataset.

Table 2: Statistics of the Italian Parkinson's and Minsk2019 ALS database used. After (Dimauro & Girardi, 2019; Vashkevich et al., 2019).

| Class → | Healthy | Pathology | |
|---|---|---|---|
| Dataset↓ | | PD | ALS |
| D1 | 220 | 220 | 77 |
| D2 | 220 | 297 | |
| D3 | - | 100 | 77 |

Table 3: Participant Distribution in the Italian Parkinson's Dataset

| Group | Subgroup | Age Range (Years) | Male (M) | Female (F) | Total |
|---|---|---|---|---|---|
| Parkinson's | Young | 19–45 | 4 | 2 | 6 |
| | Old | 50+ | 19 | 9 | 28 |
| Healthy | Young | 19–45 | 4 | 2 | 6 |
| | Old | 50+ | 10 | 12 | 22 |
| Total | | | 37 | 25 | 62 |

## 4.2 CLASSIFIER USED

The experiments were carried out using the Random Forest (RF) classifier with $100$ nestimators and random state of $42$, which is commonly used for the classification of neurodegenerative diseases. In this study, we also employ the Support Vector Machine (SVM) with *RBF* kernel and $c = 1$.
**Evaluation Metrics:** Performance of all systems is evaluated using % classification accuracy

## 4.3 FEATURE SETS USED

In this study, the performance of the proposed Constant Q Cepstral Coefficients (CQCC) and its components is compared with state-of-the-art MFCC features, as well as Jitter, Shimmer, and Teager Energy, serving as baseline features. The baseline MFCC features were extracted from the audio files at a fixed sample rate of $16\,\text{kHz}$, with a window length of $512$ samples and a window shift of $256$ samples. For the CQCC feature extraction, a minimum frequency ($f_{\min}$) of 20 Hz was set, and a total of 20 CQCC coefficients were extracted to evaluate the performance against a total of 13 MFCC coefficients.

## 5 EXPERIMENTAL DISCUSSION

### 5.1 SPECTROGRAPHIC ANALYSIS

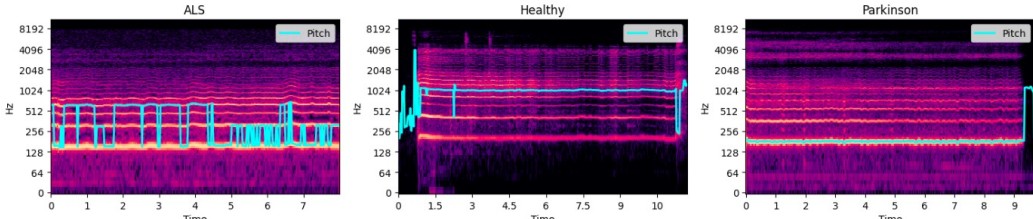

Figure 2: Spectrographic Analysis and Pitch Contours for Various Disorders Best viewed in colour.

Key regions affected include the corticospinal tract and the motor cortex. The degeneration in these areas results in impaired voluntary muscle movements, which manifest as the observed instability and interruptions in phonation in ALS patients. As observed from Figure 2(a), the spectrogram of an ALS patient demonstrates irregular and sporadic pitch contours with significant frequency fluctuations. The instability in pitch is indicative of the muscle weakness and severe effect on vocal cord control. The frequent breaks and variations in the pitch contour reflect the effortful and strained

nature of speech in ALS patients. ALS impacts the motor neurons in both the brain and spinal cord, leading to muscle weakness and spasticity. Additionally, the harmonic structure in the ALS spectrogram appears less regular and more fragmented, which mirrors the effortful and strained nature of speech. This irregularity arises from inconsistent vocal fold vibrations due to impaired muscle control. Temporal patterns in the ALS spectrogram may show uneven or interrupted speech segments, reflecting the effortful and strained nature of their speech production.

In contrast, the spectrogram of a healthy individual, as illustrated in Figure 2(b), exhibits a stable and consistent pitch contour. The formant frequencies, represented by the horizontal bands, are well-defined and continuous over time. This stability and clarity in the spectral features are typical of normal phonation, where the vocal cords vibrate regularly and smoothly, producing a steady pitch. Healthy individuals have intact motor neuron function and brain structures, which allow for precise control over the vocal apparatus. The harmonic structure in healthy individuals is regular and well-defined, indicating smooth and consistent vocal fold vibrations. Temporal patterns are regular and continuous, indicative of fluent and effortless speech.

Parkinson's disease primarily affects the substantia nigra in the basal ganglia, leading to dopamine deficiency and resulting in impaired motor control and reduced vocal cord movement. As observed from Figure 2(c), the spectrogram of a Parkinson's patient reveals a relatively stable but low-pitched contour compared to the healthy individual. The pitch contour is more monotone, reflecting the characteristic of Parkinson's disease. This monotonic pitch, along with reduced amplitude modulation, results from the reduced range and control of vocal cord movements in Parkinson's patients. The harmonic structure in Parkinson's patients may show reduced harmonic energy and lower overall intensity, reflecting the softer and more monotone speech pattern. Temporal patterns in Parkinson's patients may show prolonged phonation of certain sounds and a reduced speech rate, contributing to their overall monotone speech.

## 5.2 EXPERIMENTAL RESULTS AND DISCUSSION

This section discusses the overall performance of proposed CQCC feature against baseline features. Further, it also discusses about the spectrographic analysis between different neurodegenerative disorders and finally, LDA plots are anaysed for better feature vizulizations.

### 5.2.1 OVERALL PERFORMANCE FOR BINARY CLASSIFICATION

In this subsection, we discuss the results obtained on binary classification for healthy *vs.* pathological speech for database **D2** considered in this work. Table 4 reports the accuracy obtained on both classifiers for all the features sets considered in this study.

Table 4: Classification Accuracy of RF and SVM for Different Features

| Classifier | Jitter | Shimmer | Teager Energy | MFCC | CQCC |
|---|---|---|---|---|---|
| RF Accuracy (%) | 63.4 | 62.9 | 65.3 | 95.1 | 99.0 |
| SVM Accuracy (%) | 53.8 | 65.3 | 62.5 | 88.4 | 63.4 |

As observed from Table 4, it can be observed that, Among the features analyzed, CQCC achieved the highest classification accuracy, with the Random Forest classifier attaining an exceptional 99%, in contrast to the 63.4% accuracy achieved by the Support Vector Machine classifier. CQCC excels due to its sophisticated time-frequency representation, which captures subtle and intricate spectral variations essential for distinguishing pathological speech from healthy speech. This feature's detailed depiction of temporal and frequency characteristics enables the RF classifier to effectively discern and leverage complex patterns indicative of pathological conditions. The superior performance of RF with CQCC underscores its ability to handle and interpret the nuanced information provided by this feature. This suggests that CQCC, combined with RF's advanced classification capabilities, provides a robust framework for identifying subtle speech abnormalities with high precision.

### 5.2.2 CLASSIFICATION BETWEEN DIFFERENT PATHOLOGIES

As studied in section 5.2.1, it was observed that CQCC outperformed the MFCC, Jitter, Shimmer, and Teager energy feature sets for the classification of healthy versus pathological sounds. Here, we

Table 5: Classification Accuracy of RF and SVM for Different Features

| Classifier | Jitter | Shimmer | Teager Energy | MFCC | CQCC |
|---|---|---|---|---|---|
| RF Accuracy (%) | 41.6 | 57.6 | 49.6 | 84.6 | 90.3 |
| SVM Accuracy (%) | 44.2 | 50.9 | 44.2 | 73.0 | 80.7 |

discuss the results obtained on multiple pathological classifications. Two new databases **D1** and **D3** were prepared, where two different pathologies, along with healthy controls from both databases, were considered. Table 5 reports the results on baseline as well as proposed feature sets using RF and SVM as classifiers. It can be observed from the tables that the proposed CQCC features outperform the baseline MFCC features with an absolute increment of 5.6% and 7.7% on RF and SVM classifiers, respectively. To that effectm it can be observed that CQCC has ability to provide a comprehensive depiction of both temporal and frequency characteristics enables RF to effectively discern and leverage the complex patterns indicative of these conditions.

Table 6: Classification Accuracy of RF and SVM for Different Features

| Classifier | Jitter | Shimmer | Teager Energy | MFCC | CQCC |
|---|---|---|---|---|---|
| RF Accuracy (%) | 63.8 | 69.4 | 66.6 | 80.5 | 80.5 |
| SVM Accuracy (%) | 52.7 | 69.4 | 63.8 | 63.8 | 86.1 |

Furthermore, Table 6, shows the classification results between ALS and Parkinson's patients across different acoustical features when employing Random Forest (RF) and Support Vector Machine (SVM) classifiers. It can be observed from Table 6 that CQCC yields the highest accuracy with SVM (86.1%) and consistently performs well with RF (80.5%), indicating its superior capability in capturing the nuanced differences in the vocal characteristics associated with these diseases. On the other hand, features like Jitter and Shimmer show relatively lower accuracies, particularly with SVM (52.7% and 69.4% respectively), highlighting that these perturbation measures might not capture the disease-specific vocal characteristics as effectively. Teager Energy and MFCC (Mel Frequency Cepstral Coefficients) also show moderate performance, indicating their utility but not as robust as CQCC.

### 5.2.3 FEATURE VISULIZATION USING LDA PLOTS

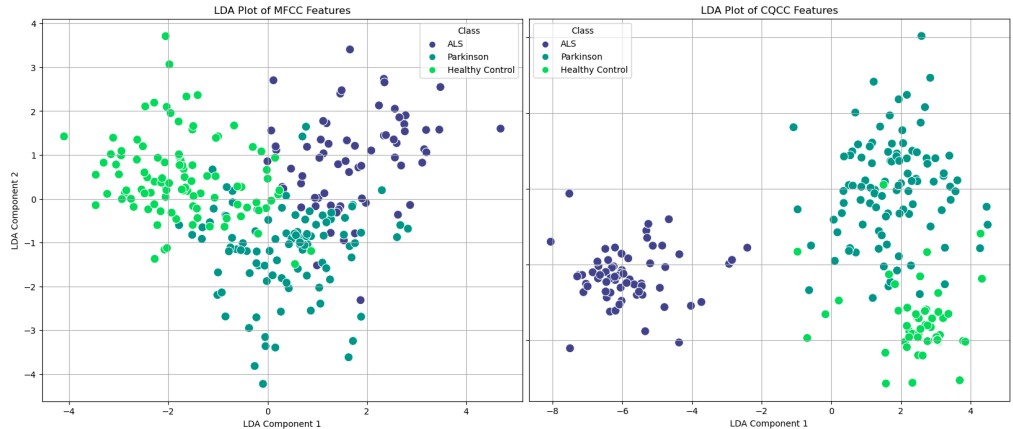

Figure 3: LDA plots for MFCC and CQCC features showing improved class separation with CQCC. Best viewed in colour.

The LDA plot as shown in Figure 3, MFCC features reveals a moderate overlap between the three classes. ALS and Parkinson's disease samples display a slight separation along the first LDA component, with Parkinson's samples tending to cluster more closely in a specific region, while ALS shows broader dispersion. Healthy Control samples, although overlapping with ALS and Parkinson,

are more distinguishable, particularly in the negative region of the first component. This moderate separability suggests that while MFCC captures useful information related to voice characteristics, it may not be fully sufficient for high-accuracy classification of the three groups. However, the LDA plot of CQCC features exhibits a clearer separation, especially between the ALS and Parkinson's disease classes. ALS samples are tightly clustered on the far left, showing a distinct separation from the Parkinson and Healthy Control classes. Healthy Control samples are spread across a different region, especially in the positive range of the first LDA component, indicating less overlap with Parkinson's disease samples. This stronger discriminative power indicates that CQCC features are more effective at distinguishing between neurodegenerative disorders and healthy individuals, making them a more robust feature set for classification tasks.

## 6 Conclusions

This study comprehensively assessed various characteristics to distinguish between healthy and pathological speech using SVM and RF classifiers. The findings underscore that CQCCs emerged as the most effective feature, achieving the highest accuracy in classification tasks. Particularly notable was RF's performance, significantly outperforming SVM, which highlights RF's capability in leveraging intricate time-frequency representations inherent in CQCC. It also demonstrated substantial efficacy, surpassing traditional measures such as Jitter, Shimmer, and Teager Energy in accuracy. This underscores the relevance of the quality features of the cepstral to accurately identify pathological speech conditions. Comparison with MFCC further validated the superiority of CQCC, showing considerable improvements in both the RF and SVM classifiers. Furthermore, the evaluation in new databases (D1 and D3) reaffirmed the robustness of CQCC in handling complex pathological classifications, providing further validation of their utility in clinical applications. Future research directions should focus on validating these findings in diverse pathological datasets and exploring advanced machine learning techniques to further improve classification precision.

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
