# OpenReview forum: "Neuroacoustic Patterns: Constant Q Cepstral Coefficients for the Classification of Neurodegenerative Disorders"
_ICLR.cc/2025/Conference — ICLR 2025 Conference Withdrawn Submission_

### Official Review · Reviewer_be3B · 2024-10-17

**Soundness:** 3
**Presentation:** 2
**Contribution:** 1
**Rating:** 1
**Confidence:** 5

**Summary:**

The paper explores the discriminatory ability of Constant Q Cepstral Coefficients (CQCC) to classify neurodegenerative disorders based on the utterance of sustained vowels. The experimental setup includes samples from patients suffering from Parkinson-s Disease (PD) and Amyotrophic Lateral Sclerosis. The proposed pipeline includes using SMOTE to compensate for class-inbalance problems and two classical ML classification models: SVM and RF. The results show a comparison between CQCC against the well-known MFCC and classical acoustic parameters related to the fundamental frequency variability.

**Strengths:**

The paper presents results demonstrating that the use of CQCC enhances classification accuracy in detecting neurodegenerative disorders compared to traditional MFCC and acoustic parameters.

**Weaknesses:**

The main drawback of the paper is its novelty. Moreover, I consider it entirely out of the Conference's scope since it does not introduce any approach incorporating the idea of a "learning representation." The components related to Machine Learning used in the paper are traditional ML models. Regarding its novelty, the set of features analysed in the paper was introduced back in 2017 and has been tested before in several voice/speech processing applications, so its contribution would be more focused on the academic community interested in the specific area of neurodegenerative disorders classification from speech signals. However, even considering the potential contribution in the area of neurodegenerative disorders detection, the comparison proposed in the paper is pretty limited since some previous works have shown that, in the context of PD detection, Rasta-PLP coefficients have better performance than MFCC but more importantly, that sustained vowels lack articulatory information which is critical to the PD detection. Indeed, there are not many datasets available out there, but the Italian Dataset used in the experiments has pretty low recording quality, and many works have shown that classifying PD vs. Control in that dataset is not a difficult task, so it should not be used as a benchmark.

**Questions:**

- Why do the authors consider the paper suitable for the ICLR venue?
- The revision of the previous work should be improved significantly; literature using spectral/cepstral features is abundant. Moreover,  to evaluate the proposed approach's actual contribution, the paper should analyse (and compare) the proposed approach with works using end-to-end approaches based on Spectrograms or feature vectors obtained from foundational models, such as Wav2vec, Speech2Vec, or HuBERT.
- Why did the authors not include experiments using Rasta-PLP if several works have reported better performance than MFCC in the context of PD detection?
- Why did the authors not include experiments using oral diadochokinesis tasks or free speech, which are currently the maximum performance tasks for PD detection from speech signals?
- Why did the authors not include more datasets in their experiments, such as the GITA (https://www5.informatik.uni-erlangen.de/fileadmin/research/Publikationen/2014/Orozco14-NSS.pdf) or Neurovoz (https://arxiv.org/abs/2403.02371) datasets, which are provided by request. There are also datasets in German, Czech, and English used in many studies that could be used by requesting the material from the authors.
- The authors should include cross-dataset experiments, which are the most challenging evaluations, where most of the proposed approaches fail or show significant drops in their performance, so they constitute the goal standard for evaluating advances in this field of application.

---

### Official Review · Reviewer_U5Xt · 2024-11-04

**Soundness:** 2
**Presentation:** 3
**Contribution:** 2
**Rating:** 3
**Confidence:** 5

**Summary:**

The article investigates constant-Q cepstral coefficients (CQCC) to perform classification of neuro-degenarative disorder from speech, and compared the results with respect to standard mel-frequency cepstral coefficients (MFCCs) and other low-level acoustic features, such as jitter, shimmer, teager-energy etc. The results presented in the paper indicate sufficient performance improvement compared to the MFCC baseline.

While this is a well motivated work that has the potential to impact detection of neuro-degenerative diseases using speech as the input modality, however it is not clear completely what the main novelty of the paper is. The authors did specify that the constant-Q cepstra is the main novelty presented in this work, however that is fairly incremental as such features have been used in speech technologies, perhaps not in the same application area as this article.

**Strengths:**

The paper focuses on speech based detection of neuro-degenerative disease, specifically Parkinson's disease and Amyotrophic lateral sclerosis (ALS). The paper is well motivated, clearly outlines the prior work that has been done and the contribution of the paper. Results presented in the article shows a strong performance demonstrated by the proposed approach as compared with MFCC-based system.

**Weaknesses:**

This is an interesting and relevant work focusing on detection/recognition of Parkinson's disease and Amyotrophic lateral sclerosis (ALS) from speech data, consisting of sustained vowels, specifically focusing on constant-Q cepstral coefficients (CQCC) as acoustic features. There are certain aspects that needs to be addressed -
(1) Given the findings are primarily based on sustained vowels, how do the observations generalize to spontaneous speech? Is it absolutely needed to have speech containing sustained vowel to be able to detect/recognize the condition investigated in this work?
(2) Table 2 in the dataset section, introduces three datasets: D1, D2 and D3. However it is not clear which one of these correspond to the datasets detailed in section 4.1. Also, in section 4.1, there are two datasets that are introduced: (a) Italian Parkinson’s Voice and Speech dataset, and (b) Minsk2019 ALS database. Table 2 is confusing as it introduces three datasets, and it is not clear what is the 3rd dataset, and which datasets correspond to D1, D2 and D3.
(3) Section 4.3 introduces MFCCs as state-of-the-art: I wonder about the rationale behind stating that MFCCs are state-of-the-art. Is there any prior work that established MFCCs as the state-of-the-art feature for this specific application?
(4) There are some typing errors that can be addressed by proof-reading the paper:
(a) page 2, section 2, line 094: "•Furthermore, no studies..." >> "• Furthermore, no studies... "
(b) page 2, section 2, line 097: "this is the first study of it;s kind ... " >> "this is the first study of it's kind ... "
(c) page 5, section 4.1, line 264: "..sustained sounds of all vowel sounds .. " > please rephrase this line, "sounds" is repeated twice and it makes the sentence a bit confusing.

**Questions:**

The paper presents an interesting and relevant application of speech technologies for detection of Parkinson's disease and Amyotrophic lateral sclerosis (ALS) from speech data, consisting of sustained vowels. Please find below some open questions, which if addressed, can facilitate the paper to be more accessible to the general reader/audience.

(1) What is meant by D1, D2 and D3 in table 2? Is it possible to specify which ones correspond to the two datasets specified earlier: (a) Italian Parkinson’s Voice and Speech dataset, and (b) Minsk2019 ALS database?

(2) The dataset section 4.1 does not provide any detail on how the train, validation and test sets are created/obtained from the data shown in tables 2 and 3. Were there any speaker overlap between the train-dev-test splits?

(3) I wonder the rationale behind stating that MFCCs are state-of-the-art. Is there any prior work that established MFCCs as the state-of-the-art feature for this specific application?

(4) It is also not clear why 20 CQC coefficients were selected against 13 MFC coefficients? What is the rationale behind using 13 MFCCs only? Typically 13 is selected for speech recognition purposes, as higher cepstral coefficients are known to capture more speaker related attributes.

(5) Section 5.1 presents an interesting analysis using some examples, however it is not clear that how much of the observations shared in the analysis is captured by the features. Specifically the first 13 cepstral features may not capture speaker specific characteristics, including pitch. It is also not obvious if harmonic energy is captured well in the explored features.
How consistent are these observations w.r.t speakers having varying degrees of ALS or Parkinsons disease?

(6) Section 5.2.2 presents an interesting analysis by comparing the findings against other relevant features. However, it will be useful to share if there is any prior art that have proposed the use of these features in isolation?
Given jitter, shimmer and teager energy features, each capture different attributes in the acoustic speech signal, these features are usually used in combination with one another, rather than isolation. It is not clear why these features were explored in isolation as baseline. What happens when these features were combined, even when combined with MFCCs of CQCCs.

(7) Section 5.2.2, page 8, lines 385-386: "Two new databases D1 and D3 were prepared, where two different pathologies .." > it is not clear what D1, D2 and D3 represent? What are the two different pathologies specified here?

(8) I am wondering if the authors have considered using some of the paralinguistic feature sets well known in the literature such as the openSMILE features that contain attributes which have been used for analysis (table 6) shared in the paper.

---

### Official Review · Reviewer_JA4h · 2024-11-04

**Soundness:** 3
**Presentation:** 2
**Contribution:** 3
**Rating:** 6
**Confidence:** 4

**Summary:**

The authors have proposed Constant Q Cepstral Coefficients (CQCC) as a measure to identify neurodegenerative diseases  (like Parkinson’s and Amyotrophic lateral sclerosis). The proposed measure is compared against basic acoustic features Jitter Shimmer Teager Energy and MFCC using traditional machine learning classifiers like random forest and Support vector machines. The discriminator power of CQCC is demonstrated using two different datasets i.e. Italian Parkinson’s Voice and Speech dataset and Minsk2019 ALS database.

**Strengths:**

1. The paper contributes towards developing interpretable features representation for neurodegenerative diseases.
2. A comparison with mostly commonly used features like MFCC
3. evaluation on two different languages and diseases.
4. achievement of significant performance over widely used feature sets by neurodegenerative research community.
5. demonstration of improved class separation of CQCC over MFCC using LDA plots.

**Weaknesses:**

1. hyperparameter optimization is not performed.
2. it is not clear what is the dimensionality of each feature set, consider adding a table or paragraph in the methodology section detailing the dimensionality of each feature set used.
3. Have you considered discussing the trade-offs between your approach and deep learning methods like wav2vec or BERT? This could help contextualize your choice of method and highlight any advantages in terms of interpretability or computational efficiency.
4. As the research field lacks large amount of datasets. In the limitations section, could you discuss how the scarcity of large datasets in this field might impact the generalizability of the findings, and what implications this has for future research?
5. Could you provide more context in the methodology section about why these specific traditional feature sets were chosen for comparison? Are there particular characteristics of these features that make them relevant benchmarks for neurodegenerative disease detection?
6. consider adding more references

**Questions:**

1) why and how your proposed feature is helpful?
2) what characteristic of speech the features are representing and how they represent neurodegeneration in speech (interpretability for clinicians?)
3) why you have not performed fusion of features?
4) any computational cost advantages?
5) explain dimensionality of feature sets and the time window for extraction of features, and how did you generate a representation for an audio recording.
6) how did you handle the variable duration of audio recordings?

---

### Official Review · Reviewer_uVwr · 2024-11-05

**Soundness:** 2
**Presentation:** 3
**Contribution:** 2
**Rating:** 5
**Confidence:** 3

**Summary:**

The paper introduces a new feature extraction method that leverages the form-invariance property of the Constant Q Transform (CQT). It is applied for the classification of neurodegenerative disorders, specifically Parkinson's Disease (PD) and Amyotrophic Lateral Sclerosis (ALS). The authors propose that CQCC, which leverages geometrically spaced frequency bins, provides superior spectrotemporal resolution compared to traditional Mel Frequency Cepstral Coefficients (MFCC). The study demonstrates that CQCC, when integrated with Random Forest and Support Vector Machine classifiers, significantly outperforms MFCC, achieving absolute improvements of 5.6% and 7.7%, respectively. The effectiveness of CQCC is validated using the Italian Parkinson’s database and the Minsk2019 database of ALS

**Strengths:**

This paper leverages the form-invariance property of the Constant Q Transform (CQT) to achieve superior spectrotemporal resolution compared to traditional Mel Frequency Cepstral Coefficients (MFCC). The authors demonstrate significant improvements in classification accuracy using Random Forest and Support Vector Machine classifiers, validated across multiple datasets. While the technical complexity may pose challenges for some readers, the research is of high quality and holds significant potential for advancing early diagnosis and treatment of neurodegenerative diseases. The paper's contributions are a valuable addition to the niche application area of disease diagnosis via audio.

**Weaknesses:**

1. Rigor in experimentation
1a. Error Analysis/ Literature comparison
 Although the paper includes spectrographic analysis and Linear Discriminant Analysis (LDA) plots to visualize feature separability, It is necessary to Conduct a detailed literature analysis of the SoTA models used for this task such as Deep feature extractors. A useful analysis would be to Identify common patterns or features that contribute to the errors and provide insights into the specific cases where the proposed method fails. This would help in understanding the contribution of CQCC as an efficient feature extractor.

1a. Cross validation/ 10 fold CV-
The paper lacks external validation of the proposed CQCC method. While simple accuracy score results are promising, additional validation using independent datasets not used in the training phase would provide stronger evidence of the method's effectiveness. This could involve cross-validation with other publicly available datasets or non intersecting splits from current dataset to achieve error estimates or uncertainty scores. *(Ref see Uncertainty Quantification of Deep Learning Models)


2. Dataset Diversity:
The study primarily uses the Italian Parkinson’s Voice and Speech dataset and the Minsk2019 ALS database. While these datasets are well-established, the paper could benefit from including more diverse datasets to ensure the generalizability of the findings. It would strengthen the validity of the results and demonstrate the robustness of the proposed method across various contexts

3 Lack of Comparison with Other Deep models/ deep feature extractors:
The paper compares CQCC primarily with traditional acoustic features like MFCC, Jitter, Shimmer, and Teager Energy. However, it does not provide a comparison with other advanced feature extraction methods or machine learning techniques that have go-to in most cases of such applications. Including such comparisons would provide a more comprehensive evaluation of the proposed method's performance and highlight its relative strengths and weaknesses

**Questions:**

Include comparisons with other deep feature extraction methods say Wav2Vec oPASE or even CNN techniques recently proposed in the literature. This would provide a more comprehensive evaluation of your method's performance and highlight its relative strengths and weaknesses.

---

### Note · Authors · 2024-11-22

I have read and agree with the venue's withdrawal policy on behalf of myself and my co-authors.